# REALTRACKER: SIMPLER AND BETTER POINT TRACKING BY PSEUDO-LABELLING REAL VIDEOS

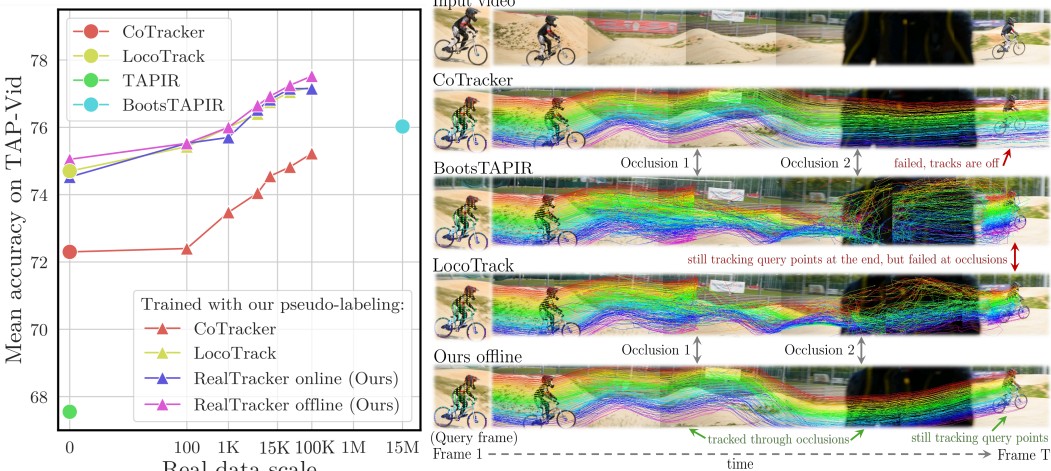

Figure 1: **Scaling point trackers using unsupervised videos.** *Left:* We compare our RealTracker, LocoTrack, CoTracker, BootsTAPIR and TAPIR. Each model is pre-trained using synthetic data (from Kubric) and then fine-tuned using real videos using our new, simple protocol for unsupervised training. Our new model and training protocol outperform SoTA by a large margin using only 0.1% of the training data. *Right:* The new model is particularly robust to occlusions.

## ABSTRACT

Most state-of-the-art point trackers are trained on synthetic data due to the difficulty of annotating real videos for this task. However, this can result in suboptimal performance due to the statistical gap between synthetic and real videos. In order to understand these issues better, we introduce RealTracker, comprising a new tracking model and a new semi-supervised training recipe. This allows real videos without annotations to be used during training by generating pseudo-labels using off-the-shelf teachers. The new model eliminates or simplifies components from previous trackers, resulting in a simpler and often smaller architecture. This training scheme is much simpler than prior work and achieves better results using 1,000 times less data. We further study the scaling behaviour to understand the impact of using more real unsupervised data in point tracking. The model is available in online and offline variants and reliably tracks visible and occluded points.

## 1 INTRODUCTION

Tracking points is a key step in the analysis of videos, particularly for tasks like 3D reconstruction and video editing that require precise recovery of correspondences. Point trackers have evolved significantly in recent years, with designs based on transformer neural networks inspired by PIPs (Harley et al., 2022). Notable examples include TAP-Vid (Doersch et al., 2022), which introduced a new benchmark for point tracking, and TAPIR (Doersch et al., 2023), which introduced an improved tracker that extends PIPs' design with a global matching stage. CoTracker (Karaev et al.,

2024b) proposed a transformer architecture that tracks multiple points jointly, with further gains in tracking quality, particularly for points partially occluded in the video.

In this paper, we introduce a new point tracking model, *RealTracker*, that builds on the ideas of recent trackers but is significantly simpler, more data efficient, and more flexible. Our architecture, in particular, removes some components that recent trackers proposed as necessary for good performance while still improving on the state-of-the-art. For the first time, we also investigate the data scaling behaviour of a point tracker and show the advantages of different model architectures and training protocols in terms of final tracking quality and data efficiency.

The excellent performance of recent trackers is due to the ability of high-capacity neural networks to learn a robust prior from many training videos and use this prior for tackling complex and ambiguous tracking cases, such as occlusions and fast motion. Therefore, the availability of high-quality training data (Muller et al., 2018) is of crucial importance in obtaining solid tracking results.

While, in principle, there is no shortage of videos that could be used to train point trackers, it is difficult to manually annotate them with point tracks (Doersch et al., 2022). Fortunately, synthetic videos (Greff et al., 2022), which can be annotated automatically, have been found to be a good substitute for real data for low-level tasks like point tracking (Harley et al., 2022). Still, a diverse collection of synthetic videos is expensive at scale, and the sim-to-real gap is not entirely negligible. Hence, using real videos to train point trackers remains an attractive option.

Recent works have thus explored utilizing large collections of real but unlabelled videos to train point trackers. BootsTAPIR (Doersch et al., 2024), in particular, has recently achieved state-of-the-art accuracy on the TAP-Vid benchmark by training a model on 15 million unlabelled videos. While the benefits of using more training data have thus been demonstrated, the data scaling behaviour of point trackers is not well understood. In particular, it is unclear if the millions of real training videos used in BootsTAPIR are necessary to train a good tracker. The same can be said about the benefits of their relatively complex semi-supervised training recipe.

Another largely unexplored aspect is the competing designs of different trackers. Transformer architectures like PIPs (Harley et al., 2022), TAPIR (Doersch et al., 2023), and CoTracker (Karaev et al., 2024b), as well as more recent contributions like LocoTrack (Cho et al., 2024), propose each significant changes, extensions, new components, and different design decisions. While these are shown to help in the respective papers, it is less clear if they are all essential or whether these designs can be simplified and made more efficient.

RealTracker contributes to answering these questions. Our model is based on a simpler architecture and training protocols than recent trackers such as BootsTAPIR and LocoTrack. It outperforms BootsTAPIR by a significant margin on the TAP-Vid and Dynamic Replica (Karaev et al., 2023) benchmarks while using *three orders of magnitude fewer unlabelled videos* and a simpler training protocol than BootsTAPIR. We also study the data scaling behaviour of this model under increasingly more real training videos. LocoTrack benefits in a similar manner to RealTracker from scaling data but cannot track occluded points well.

RealTracker borrows elements from prior models, including iterative updates and convolutional features from PIPs, cross-track attention for joint tracking, virtual tracks for efficiency, and unrolled training for windowed operation from CoTracker, as well as the 4D correlation from LocoTrack. At the same time, it significantly simplifies some of these components and removes others, such as the global matching stage of BootsTAPIR and LocoTrack. This helps to identify which components are really important for a good tracker. RealTracker's architecture is also flexible as it can operate both offline (i.e., single window) and online (i.e., sliding window) if trained in the same way.

## 2 RELATED WORK

**Tracking-Any-Point.** The task of *tracking any point* was introduced by PIPs (Harley et al., 2022), who revisited the classic Particle Video (Sand & Teller, 2008) method and proposed to use deep learning for point tracking. Inspired by RAFT (Teed & Deng, 2020), an optical flow algorithm, PIPs extracts correlation maps between frames and feeds them into a network to refine the track estimates. TAP-Vid (Doersch et al., 2022) improved the framing of the problem and proposed three different benchmarks for point tracking. TAPIR (Doersch et al., 2023) combined TAP-Vid-like global match-

ing with PIPs, resulting in a much-improved performance. (Zheng et al., 2023) introduced another synthetic benchmark, PointOdyssey, and PIPs++, an improved version of PIPs that can track points over extended durations. CoTracker (Karaev et al., 2024b) noted that a strong correlation exists between different tracks, which can be exploited to improve tracking, particularly behind occlusions and out-of-frame. Le Moing et al. (2024) further improved CoTracker by densifying its output. VGGSfM (Wang et al., 2024) proposed a coarse-to-fine tracker design where tracks are validated through 3D reconstruction, but it only targets static scenes. Inspired by DETR (Carion et al., 2020), Li et al. (2024) introduced TAPTR, an end-to-end transformer architecture for point tracking, representing points as queries in the transformer decoder. LocoTrack (Cho et al., 2024) extended 2D correlation features to 4D correlation volumes while also simplifying the point-tracking pipeline and making it more efficient. Our work proposes a further simplified framework that runs 27% faster than LocoTrack, maintaining the ability to track occluded points via joint tracking, like CoTracker.

Annotating data for point tracking is particularly challenging due to the required precision: the annotation should have (at least) pixel-level accuracy. The prevailing paradigm in point tracking is thus to train models using synthetic data, where such annotations can be obtained automatically and without errors, and show that the resulting models generalize to real data. All the methods mentioned above follow this paradigm, and most are trained solely on Kubric (Greff et al., 2022).

**Semi-supervised correspondence.** An alternative to synthetic data is unlabelled real data in combination with unsupervised or semi-supervised learning. For example, one can use photometric consistency as a proxy for correspondences. Such training is well suited for optical flow and dense tracking but often leads to false matches due to occlusions, repeated textures, or lighting changes. Therefore it usually requires multi-frame estimates (Janai et al., 2018), explicit reasoning about occlusions (Wang et al., 2018), hand-crafted loss terms (Liu et al., 2019b; Meister et al., 2018), or various data augmentation strategies (Liu et al., 2020). Alternatively, one can use an existing tracker to train another in a process akin to distillation (Liu et al., 2019a). More robust unsupervised learning signals for long-range tracking can be obtained via simple colorization of gray-scale videos by copying colors from the reference frame (Vondrick et al., 2018) or utilizing richer visual patterns (Lai et al., 2020).

Accounting for cycle consistency (Wang et al., 2019; Jabri et al., 2020) or temporal continuity (Földiák, 1991; Wiskott & Sejnowski, 2002) in videos is another way to obtain a reliable proxy signal to learn correspondences without full supervision, or even learn generic visual features (Goroshin et al., 2015; Wang & Gupta, 2015). Shen et al. (2022) proposed an unsupervised learning framework based on siamese networks for training trackers with cycle consistency. Recently, (Sun et al., 2024) proposed refining PIPs and RAFT on a pre-generated dataset with pseudo-labels using color constancy and cycle consistency signals. This pipeline improves tracking, but performance quickly saturates. Wu et al. (2021) introduced an unsupervised learning framework that does not require any annotated videos in visual tracking. Tumanyan et al. (2024) combined test-time per-video optimization with DINOv2 (Oquab et al., 2023) features to improve point tracking. Karaev et al. (2024a) introduced a more efficient architecture and a simple pseudo-labelling pipeline to further improve its performance by training on real data.

Most relevant to our work, BootsTAPIR (Doersch et al., 2024) improved TAPIR trained on Kubric by fine-tuning it on 15 million real videos using self-training while retaining a small synthetic dataset with ground-truth supervision to avoid catastrophic forgetting. They proposed applying augmentations to student predictions and trained the model with an exponential moving average (EMA) while computing three different loss masks for robustness. In contrast, our approach uses a simpler design which does not require augmentations, masks, or EMA for training. We also do not need ground-truth supervised data during finetuning on pseudo-labels. Instead, our idea is to train a student model by utilizing existing trackers with complementary qualities as teachers. We also show that this protocol only requires a small fraction of the real videos utilized in BootsTAPIR.

## 3 METHOD

In this section, we formally introduce the task of point tracking and then outline the proposed Real-Tracker architecture and the pseudo-labelling training pipeline we use to train it.

Given a video $(\mathcal{I}_t)_{t=1}^{T}$, which is a sequence of $T$ frames $\mathcal{I}_t \in \mathbb{R}^{3 \times H \times W}$, and a query point $\mathcal{Q} = (\mathbf{t}^q, \mathbf{x}^q, \mathbf{y}^q) \in \mathbb{R}^3$ where $\mathbf{t}^q$ indicates the query frame index and $(\mathbf{x}^q, \mathbf{y}^q)$ represents the initial location of the query point, our goal is to predict the corresponding point track $\mathcal{P}_t = (\mathbf{x}_t, \mathbf{y}_t) \in \mathbb{R}^2$, $t = 1, \ldots, T$, with $(\mathbf{x}_{\mathbf{t}^q}, \mathbf{y}_{\mathbf{t}^q}) = (\mathbf{x}^q, \mathbf{y}^q)$. As is common in modern point tracking models (Doersch et al., 2023; Karaev et al., 2024b), RealTracker also estimates visibility $\mathcal{V}_t \in [0, 1]$ and confidence $\mathcal{C}_t \in [0, 1]$. Visibility shows whether the tracked point is visible ($\mathcal{V}_t = 1$) or occluded ($\mathcal{V}_t = 0$) in the current frame, while confidence measures whether the network is confident that the tracked point is within a certain distance from the ground truth in the current frame ($\mathcal{C}_t = 1$). The model initializes all tracks with query coordinates $\mathcal{P}_t := (\mathbf{x}_{\mathbf{t}^q}, \mathbf{y}_{\mathbf{t}^q}), t = 1, \ldots, T$, confidence and visibility with zeros $\mathcal{C}_t := 0, \mathcal{V}_t := 0$, then updates all of them iteratively.

### 3.1 TRAINING USING UNLABELLED VIDEOS

Recent trackers are trained primarily on synthetic data (Greff et al., 2022) due to the challenge of annotating real data for this problem at scale. However, BootsTAPIR (Doersch et al., 2024) has shown that it is possible to train better trackers by adding to the mix unlabelled real videos. In order to do so, they propose a sophisticated self-training protocol that uses a large number of unlabelled videos (15M), self-training, data augmentations, and transformation equivariance.

Here, we propose a much simpler protocol that allows us to surpass the performance of (Doersch et al., 2024) with 1,000× less data: we use *a variety of existing* trackers to label a collection of real videos, using them as *teachers*, and then use the pseudo-labels to train a new *student* model, which we pre-train utilizing synthetic data.

Importantly, the teacher models are *also* trained using the same synthetic data only. One may thus wonder why this protocol should result in a student being better than any of the teachers. There are several reasons for this: (1) the student benefits from learning from a much larger (noisy) dataset than the synthetic data alone; (2) learning from real videos mitigates the distribution shifts between synthetic and real data; (3) there is an ensembling/voting effect which reduces the pseudo-annotations noise; (4) the student model may inherit the strengths of the different teachers, which may excel in different aspects of the task (e.g., offline trackers track occluded points better and online trackers tend to stick to the query points more closely near the track's origin).

**Dataset.** In order to enable such training, we collected a large-scale dataset of Internet-like videos (around 100,000 videos of 30 seconds each) featuring diverse scenes and dynamic objects, primarily humans and animals. We demonstrate that performance improves when training on increasingly larger subsets of this data, starting from as few as 100 videos (see Figure 1).

**Teacher models.** To create a diverse set of supervisory signals, we employ multiple teacher models trained only on synthetic data from Kubric (Greff et al., 2022). Our set of teachers consists of our proposed models RealTracker online and RealTracker offline, CoTracker (Karaev et al., 2024b), and TAPIR (Doersch et al., 2023). During training, we randomly and uniformly sample a frozen teacher model for every batch (meaning that it is likely that, over several epochs, the same video will receive pseudo-labels from different teachers), which helps to prevent over-fitting and promotes generalization. The teacher models are not updated during training.

**Query point sampling.** Trackers require a query point to track in addition to a video. After randomly choosing a teacher for the current batch, we sample a set of query points for each video. To select such queries, we use the SIFT detector (Lowe, 1999) sampling, biasing the selection of points to those which are "good to track" (Shi & Tomasi, 1994). Specifically, we randomly select $\hat{T}$ frames across a video and apply SIFT to generate points to start tracks on these keyframes. Our intuition behind using a feature extractor is guided by its ability to detect descriptive image features whenever possible while failing to do so when meeting ambiguous cases. We hypothesise that this will serve as a filter for hard-to-track points and will thus improve the stability of training. Following this intuition, if SIFT fails to produce a sufficient number of points for any frame, we skip the video completely during training to maintain the quality of our training data.

**Supervision.** We supervise tracks predicted by the student model with the same loss used to pre-train the model on synthetic data, with only minor modifications for handling occlusion and tracking confidence. These details are given later in Section 3.3.

Figure 2: **Architecture.** We compute convolutional features for every frame of the given video, and then the correlations between the feature sampled around the query frame for the query point and all the other frames. We then iteratively update tracks $\mathcal{P}^{(m)} = \mathcal{P}^{(m)} + \Delta \mathcal{P}^{(m+1)}$, confidence $\mathcal{C}^{(m)}$, and visibility $\mathcal{V}^{(m)}$ with a transformer that takes the previous estimates $\mathcal{P}^{(m)}, \mathcal{C}^{(m)}, \mathcal{V}^{(m)}$ as input.

### 3.2 REALTRACKER MODEL

We provide two model versions of RealTracker: offline and online. The online version operates in a sliding window manner, processing the input video sequentially and tracking points forward-only. In contrast, the offline version processes the entire video as a single sliding window, enabling point tracking in both forward and backward directions. The offline version tracks occluded points better and also improves the long-term tracking of visible points. However, the maximum number of tracked frames is memory-bound, while the online version can track in real-time indefinitely.

**Feature maps.** We start by computing dense $d$-dimensional feature maps with a convolutional neural network for each video frame, i.e., $\Phi_t = \Phi(\mathcal{I}_t), t = 1, \ldots, T$. We downsample the input video by a factor of $k = 4$ for efficiency so that $\Phi_t \in \mathbb{R}^{d \times \frac{H}{k} \times \frac{W}{k}}$, and compute the feature maps at $S = 4$ different scales, i.e., $\Phi_t^s \in \mathbb{R}^{d \times \frac{H}{k2^{s-1}} \times \frac{W}{k2^{s-1}}}$, $s = 1, \ldots, S$.

**4D correlation features.** In order to allow the network to locate the query point $\mathcal{Q} = (\mathbf{t}^q, \mathbf{x}^q, \mathbf{y}^q)$ in frames $t = 1, \ldots, T$, we compute the correlation between the feature vectors extracted from the map $\Phi_{\mathbf{t}^q}$ at the query frame $t^q$ around the query coordinates $(\mathbf{x}^q, \mathbf{y}^q)$ and feature vectors extracted from maps $\Phi_t, t = 1, \ldots, T$ around current track estimates $\mathcal{P}_t = (\mathbf{x}_t, \mathbf{y}_t)$ at the other frames.

More specifically, every point $\mathcal{P}_t$ is described by extracting a square neighbourhood of feature vectors at different scales. We denote this collection of feature vectors as:

$$\phi_t^s = \left[ \Phi_t^s \left( \frac{\mathbf{x}}{ks} + \delta, \frac{\mathbf{y}}{ks} + \delta \right) : \delta \in \mathbb{Z}, \|\delta\|_\infty \leq \Delta \right] \in \mathbb{R}^{d \times (2\Delta+1)^2}, \quad s = 1, \ldots, S, \quad (1)$$

where the feature map $\Phi_t^s$ is sampled using bilinear interpolation around the point $(\mathbf{x}_t, \mathbf{y}_t)$. Therefore, for each scale $s$, $\phi_t^s$ contains a grid of $(2\Delta + 1)^2$ pointwise $d$-dimensional features.

Next, we define the *4D correlation* (Cho et al., 2024) $\langle \phi_{\mathbf{t}^q}^s, \phi_t^s \rangle = \text{stack}((\phi_{\mathbf{t}^q}^s)^\top \phi_t^s) \in \mathbb{R}^{(2\Delta+1)^4}$ for every scale $s = 1, \ldots, S$. Intuitively, this operation compares each feature vector around the query point $(\mathbf{x}^q, \mathbf{y}^q)$ to each feature vector around the track point $(\mathbf{x}_t, \mathbf{y}_t)$, which the network uses to predict the track update. Before passing them to the transformer, we project these correlations with a multi-layer perceptron (MLP) to reduce their dimensionality, defining the *correlation features* to be: $\text{Corr}_t = \left( \text{MLP}(\langle \phi_{\mathbf{t}^q}^1, \phi_t^1 \rangle), \ldots, \text{MLP}(\langle \phi_{\mathbf{t}^q}^S, \phi_t^S \rangle) \right) \in \mathbb{R}^{pS}$, where $p$ is the projection dimension. This MLP architecture is much simpler than the ad-hoc module used by LocoTrack (Cho et al., 2024) for computing their correlation features.

**Iterative updates.** We initialize the confidence $\mathcal{C}_t$ and visibility $\mathcal{V}_t$ with zeros, and the tracks $\mathcal{P}_t$ for all the times $t = 1, \ldots, T$ with the initial coordinates from the query point $\mathcal{Q}$. We then iteratively update all these quantities with a transformer.

At every iteration, we embed the tracks using the Fourier Encoding of the per-frame displacements, i.e., $\eta_{t \to t+1} = \eta(\mathcal{P}_{t+1} - \mathcal{P}_t).$, Then, we concatenate the track embeddings (in both directions

$\eta_{t \to t+1}$ and $\eta_{t-1 \to t}$), confidence $\mathcal{C}_t$, visibility $\mathcal{V}_t$, and the 4D correlations $\mathrm{Corr}_t$ for every query point $i = 1, \ldots, N$: $\mathcal{G}_t^i = \left( \eta_{t-1 \to t}^i, \eta_{t \to t+1}^i, \mathcal{C}_t^i, \mathcal{V}_t^i, \mathrm{Corr}_t^i \right)$. $\mathcal{G}_t^i$ forms a grid of input tokens for the transformer that span time $T$ and the number of query points $N$. The transformer $\Psi$ takes this grid as input, adds standard Fourier time embeddings, and applies factorized time attention with $t = 1, \ldots, T$ and group attention with $i = 1, \ldots, N$. It also uses proxy tokens (Karaev et al., 2024b) for efficiency. This transformer estimates the updates to tracks, confidence, and visibility incrementally as $(\Delta\mathcal{P}, \Delta\mathcal{C}, \Delta\mathcal{V}) = \Psi(\mathcal{G})$. We update tracks $\mathcal{P}$, confidence $\mathcal{C}$ and visibility $\mathcal{V}$ $M$ times, where: $\mathcal{P}^{(m+1)} = \mathcal{P}^{(m)} + \Delta\mathcal{P}^{(m+1)}; \mathcal{C}^{(m+1)} = \mathcal{C}^{(m)} + \Delta\mathcal{C}^{(m+1)}; \mathcal{V}^{(m+1)} = \mathcal{V}^{(m)} + \Delta\mathcal{V}^{(m+1)}$. Note that we resample the pointwise features $\phi$ around updated tracks $\mathcal{P}^{(m+1)}$ and recompute the correlations $\mathrm{Corr}$ after every update.

## 3.3 Model training

We supervise both visible and occluded tracks using the Huber loss with a threshold of 6 and exponentially increasing weights. We assign a smaller weight to the loss term for occluded points:

$$\mathcal{L}_{\mathrm{track}}(\mathcal{P}, \mathcal{P}^\star) = \sum_{m=1}^{M} \gamma^{M-m} (\mathbb{1}_{occ}/5 + \mathbb{1}_{vis}) \, \mathrm{Huber}(\mathcal{P}^{(m)}, \mathcal{P}^\star), \qquad (2)$$

where $\gamma = 0.8$ is a discount factor. This prioritises tracking well the visible points.

Confidence and visibility are supervised with a Binary Cross Entropy (BCE) loss at every iterative update. The ground truth for confidence is defined by an indicator function that checks whether the predicted track is within 12 pixels of the ground truth track for the current update. We apply the sigmoid function to the predicted confidence and visibility before computing the loss:

$$\mathcal{L}_{\mathrm{conf}}(\mathcal{C}, \mathcal{P}, \mathcal{P}^\star) = \sum_{m=1}^{M} \gamma^{M-m} \, \mathrm{CE} \left( \sigma(\mathcal{C}^{(m)}), \mathbb{1}\left[\|\mathcal{P}^{(m)} - \mathcal{P}^\star\|_2 < 12\right]\right), \qquad (3)$$

$$\mathcal{L}_{\mathrm{occl}}(\mathcal{V}, \mathcal{V}^\star) = \sum_{m=1}^{M} \gamma^{M-m} \, \mathrm{CE}(\sigma(\mathcal{V}^{(m)}), \mathcal{V}^\star). \qquad (4)$$

**Training using pseudo-labels.** When using pseudo-labelled videos, we supervise RealTracker using the same loss (2) used for the synthetic data, but found it more stable not to supervise confidence and visibility. To avoid forgetting the latter predictions, we use a separate linear layer to estimate confidence and visibility and simply freeze it at this training stage.

**Online model.** Both online and offline versions of RealTracker have the same architecture. The main difference between them is the way of training. The online version processes videos in a windowed manner: it takes $T'$ frames as input, predicts tracks for them, then moves forward by $T'/2$ frames, and repeats this process. It uses the overlapped predictions for the tracks, confidence, and visibility from the previous sliding window as initialization for the current window.

During training, we compute the same losses (2) to (4) for the online version separately for each sliding window. Then, we take the mean across all the sliding windows. Since the online version can track points only forward in time, we compute the losses only starting from the first window with the query frame $\mathbf{t}^q$ onwards. For the offline version, however, we compute the losses for every frame because it tracks points in both directions. We train the online version on videos of the same length, while the offline version needs to see videos of different lengths during training to avoid overfitting to a specific length. With this intuition in mind, for the offline version, we randomly trim a video between $T/2$ and $T$ frames and linearly interpolate time embeddings during training.

## 3.4 Discussion

Our model includes several simplifications and improvements compared to previous architectures like PIPs, TAPIR and CoTracker. In particular: (1) The model uses the idea of 4D correlation from LocoTrack but is further simplified by utilizing a simple MLP to process the correlation features instead of their ad-hoc architecture; (2) It estimates confidence for every tracked point; (3) Compared to CoTracker, the grid of tokens $\mathcal{G}$ is simplified, using only correlation features and Fourier

| Method | Train | Kinetics | | | RGB-S | | | DAVIS | | | Mean |
|---|---|---|---|---|---|---|---|---|---|---|---|
| | | AJ ↑ | $\delta_{avg}^{vis}$ ↑ | OA ↑ | AJ ↑ | $\delta_{avg}^{vis}$ ↑ | OA ↑ | AJ ↑ | $\delta_{avg}^{vis}$ ↑ | OA↑ | $\delta_{avg}^{vis}$ ↑ |
| PIPs++ (Zheng et al., 2023) | PO | — | 63.5 | — | — | 58.5 | — | — | 73.7 | — | 65.2 |
| TAPIR (Doersch et al., 2023) | Kub | 49.6 | 64.2 | 85.0 | 55.5 | 69.7 | 88.0 | 56.2 | 70.0 | 86.5 | 68.0 |
| CoTracker (Karaev et al., 2024b) | Kub | 49.6 | 64.3 | 83.3 | 67.4 | 78.9 | 85.2 | 61.8 | 76.1 | 88.3 | 73.1 |
| TAPTR (Li et al., 2024) | Kub | 49.0 | 64.4 | 85.2 | 60.8 | 76.2 | 87.0 | 63.0 | 76.1 | 91.1 | 72.2 |
| LocoTrack (Cho et al., 2024) | Kub | 52.9 | 66.8 | 85.3 | 69.7 | 83.2 | 89.5 | 62.9 | 75.3 | 87.2 | 75.1 |
| RealTracker (Ours, online) | Kub | 54.1 | 66.6 | 87.1 | 71.1 | 81.9 | 90.3 | **64.5** | 76.7 | 89.7 | 75.1 |
| RealTracker (Ours, offline) | Kub | 53.5 | 66.5 | 86.4 | 74.0 | 84.9 | 90.5 | 63.3 | 76.2 | 88.0 | 75.9 |
| BootsTAPIR (Doersch et al., 2024) | Kub+15**M** | 54.6 | 68.4 | 86.5 | 70.8 | 83.0 | 89.9 | 61.4 | 73.6 | 88.7 | 75.0 |
| RealTracker (Ours, online) | Kub+15k | **55.8** | **68.5** | **88.3** | 71.7 | 83.6 | 91.1 | 63.8 | 76.3 | 90.2 | 76.1 |
| RealTracker (Ours, offline) | Kub+15k | 54.7 | 67.8 | 87.4 | **74.3** | **85.2** | **92.4** | 64.4 | **76.9** | **91.2** | **76.6** |

Table 1: **TAP-Vid benchmarks** RealTracker trained on synthetic Kubric shows strong performance compared to other models, while the online version fine-tuned on 15k additional real videos (Kub+15k) outperforms all the other methods, even BootsTAPIR trained on $1,000\times$ more real videos. Training data: (Kub) Kubric (Greff et al., 2022), (PO) Point Odyssey (Zheng et al., 2023).

embeddings of displacements; (4) The visibility flags are updated at each iteration along with other quantities instead of using a separate network. (5) Compared to TAPIR, BootsTAPIR and Loco-Track, RealTracker *does not* use a global matching module as we found it redundant.

A benefit of these simplifications is that RealTracker is considerably leaner and faster than other similar trackers. Specifically, RealTracker has $2\times$ fewer parameters than CoTracker, while the absence of global matching and the use of an MLP to process correlations makes RealTracker 27% faster than the fastest tracker (LocoTrack) despite cross-track attention.

## 4 EXPERIMENTS

In this section, we describe our evaluation protocol. Then, we compare our online and offline models to state-of-the-art trackers (Section 4.1), analyse their performance for occluded points (Section 4.1), show how different models scale with the proposed pseudo-labeling pipeline (Section 4.2), and ablate the design choices of the architecture and the scaling pipeline (Section 4.3).

**Evaluation protocol.** We conduct our evaluation on **TAP-Vid** (Doersch et al., 2022) comprising TAP-Vid-Kinetics, TAP-Vid-DAVIS and RGB-Stacking. TAP-Vid-Kinetics consists of 1,144 YouTube videos from the Kinetics-700–2020 validation set (Carreira & Zisserman, 2017), featuring complex camera motion and cluttered backgrounds, with an average of 26 tracks per video. TAP-Vid-DAVIS comprises 30 real-world videos from the DAVIS 2017 validation set (Perazzi et al., 2016), with an average of 22 tracks per video. RGB-Stacking is a synthetically generated dataset of robotic videos with many texture-less regions that are difficult to track.

We use the standard TAP-Vid metrics: Occlusion Accuracy (OA; accuracy of occlusion prediction as binary classification), $\delta_{avg}^{vis}$ (fraction of visible points tracked within 1, 2, 4, 8 and 16 pixels, averaged over thresholds) and Average Jaccard (AJ, measuring tracking and occlusion prediction accuracy together). All videos are resized to 256×256 pixels before being processed by the model.

Similarly, we evaluate RealTracker on **RoboTAP** (Vecerik et al., 2023), which contains 265 real-world videos of robotic manipulation tasks, with an average duration of 272 frames. Following (Doersch et al., 2022), we evaluate TAP-Vid and RoboTAP in the "first query" mode: sampling query points from the first frame where they become visible. Additionally, we also evaluate on **DynamicReplica** (Karaev et al., 2023) following (Karaev et al., 2024b). Because this dataset is synthetic, the tracker can be evaluated on occluded points. The evaluation subset of Dynamic Replica consists of 20 long (300 frames) sequences of articulated 3D models. We evaluate these benchmarks at their native resolution but resize the predictions to a resolution of 256×256 pixels and report the accuracy of visible ($\delta_{avg}^{vis}$) and occluded points ($\delta_{avg}^{occ}$) using the same thresholds as in TAP-Vid.

| Method | Train | Size↓ | Time↓ | Dynamic Replica | | RoboTAP | | | Mean |
| --- | --- | --- | --- | --- | --- | --- | --- | --- | --- |
| | | | | $\delta_{avg}^{vis}$ ↑ | $\delta_{avg}^{occ}$ ↑ | AJ ↑ | $\delta_{avg}^{vis}$ ↑ | OA↑ | $\delta_{avg}^{vis}$ ↑ |
| PIPs++ (Zheng et al., 2023) | PO | 25M | - | 64.0 | 28.5 | — | 63.0 | — | 63.5 |
| TAPIR (Doersch et al., 2023) | Kub | 31M | 293 | 66.1 | 27.2 | 59.6 | 73.4 | 87.0 | 69.8 |
| CoTracker (Karaev et al., 2024b) | Kub | 45M | 472 | 68.9 | 37.6 | 58.6 | 70.6 | 87.0 | 69.8 |
| TAPTR (Li et al., 2024) | Kub | - | - | 69.5 | 34.1 | 60.1 | 75.3 | 86.9 | 72.4 |
| LocoTrack (Cho et al., 2024) | Kub | **12M** | 290 | 71.4 | 29.8 | 62.3 | 76.2 | 87.1 | 73.8 |
| RealTracker (Ours, online) | Kub | 25M | 405 | 72.9 | 41.0 | 60.8 | 73.7 | 87.1 | 73.3 |
| RealTracker (Ours, offline) | Kub | 25M | **209** | 69.8 | 41.8 | 59.9 | 73.4 | 87.1 | 71.6 |
| BootsTAPIR (Doersch et al., 2024) | Kub+15**M** | 78M | 303 | 69.0 | 28.0 | 64.9 | **80.1** | 86.3 | 74.6 |
| RealTracker (Ours, online) | Kub+15k | 25M | 405 | **73.3** | 40.1 | **66.4** | 78.8 | **90.8** | **76.1** |
| RealTracker (Ours, offline) | Kub+15k | 25M | **209** | 72.2 | **42.3** | 64.7 | 78.0 | 89.4 | 75.1 |

Table 2: **Results on Dynamic Replica and RoboTAP.** Our approach consistently shows better results. Only $\delta_{avg}^{vis}$ on RoboTAP is better for BootsTAPIR, trained on 1,000× more data. Size in number of params; speed expressed as $\mu s$ per frame and per tracked point.

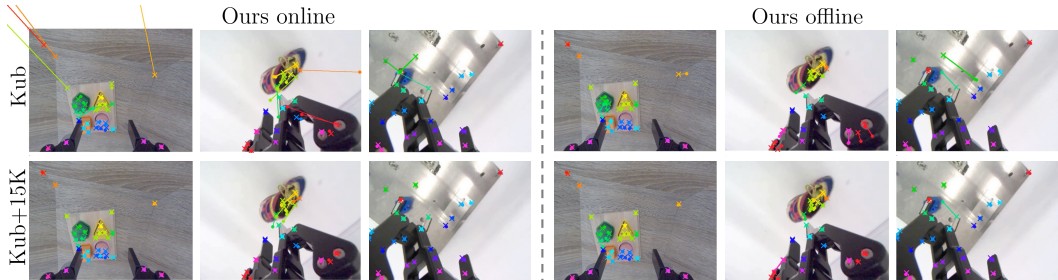

Figure 3: **Ours.** Predictions of online (first three columns) and offline (last three columns) models on RoboTAP before (first row) and after (second row) scaling. We visualize the distance between ground truth (crosses) and predictions (points). Scaling improves both online and offline models.

## 4.1 COMPARISON TO THE STATE-OF-THE-ART

For fairness with trackers blind to the correlation between different tracks, we evaluate RealTracker on TAP-Vid on one query point at a time and sample additional support points to leverage joint tracking (Karaev et al., 2024b). This ensures that no information about objects in the videos leaks to the tracker through the selection of benchmark points (which generally correlate with objects in benchmarks). We multiply predicted visibility by predicted confidence and apply a threshold to the resulting quantity as in (Doersch et al., 2023), improving the AJ and OA metrics.

As shown in Table 1, RealTracker is highly competitive with other trackers across various benchmarks even when only trained using synthetic data (Kub). Adding unlabelled videos utilizing the approach of Section 4.2 (+15k) boosts the results well above the state-of-the-art for all metrics for DAVIS, RGB-S, and Kinetics, and for two out of three metrics (AJ and OA) on RoboTAP (Table 2). The +15k offline version is even better than the online one on DAVIS and RGB-S, but worse on Kinetics and RoboTAP. As for data efficiency, despite being trained on just 15k additional real videos, our models outperform BootsTAPIR, which was trained using 15M videos (i.e., **1,000** more). Slightly better performance can be obtained by increasing the data further (Section 4.2). LocoTrack also benefits similarly from our training scheme but struggles during occlusions, as shown next.

**Tracking occluded points** We compare RealTracker with other methods on Dynamic Replica in Table 2 ($\delta_{avg}^{occ}$ and OA columns). On this benchmark, RealTracker online is better than all the other methods even when trained solely on Kubric; in particular, it is much better than LocoTrack, which justifies the additional parameters in the cross-track attention modules. Adding the 15k real videos improves the tracking of visible points for the online and offline versions, but only the offline model shows improvement in tracking occluded points. In addition to improving more, RealTracker offline tracks occluded points better than the online version. This is because accessing all video frames at once helps to interpolate trajectories behind occlusions.

| Cross-track attention | Dynamic Replica | |
|:---:|:---:|:---:|
| | $\delta_{\text{avg}}^{\text{vis}} \uparrow$ | $\delta_{\text{avg}}^{\text{occ}} \uparrow$ |
| ✗ | 71.3 | 35.9 |
| ✓ | **72.9** | **41.0** |

Table 3: **Impact of cross-track attention on occluded tracking.** Cross-track attention improves the tracking of occluded points substantially. It also improves visible points, but the effect is smaller.

| Self-training | Mean on TAP-Vid | | |
|:---:|:---:|:---:|:---:|
| | AJ↑ | $\delta_{\text{avg}} \uparrow$ | OA↑ |
| ✗ | 62.2 | 74.5 | 88.2 |
| ✓ | **63.5** | **75.7** | **89.5** |

Table 4: **Self-training.** Training Real-Tracker online on its own predictions improves the model. We use 10k real videos and train to convergence.

| RT onl. | RT offl. | TAPIR | CoTr. | Mean on TAP-Vid | | |
|:---:|:---:|:---:|:---:|:---:|:---:|:---:|
| | | | | AJ↑ | $\delta_{\text{avg}} \uparrow$ | OA↑ |
| ✗ | ✗ | ✗ | ✗ | 62.2 | 74.5 | 88.2 |
| ✓ | ✗ | ✗ | ✗ | 63.5 | 75.7 | 89.5 |
| ✓ | ✓ | ✗ | ✗ | **64.5** | 76.4 | 89.9 |
| ✓ | ✗ | ✓ | ✗ | 63.6 | 76.2 | 89.7 |
| ✓ | ✗ | ✗ | ✓ | 64.2 | 76.5 | 90.1 |
| ✓ | ✓ | ✓ | ✗ | 64.0 | 76.6 | 89.9 |
| ✓ | ✗ | ✓ | ✓ | 64.2 | 76.6 | 90.1 |
| ✓ | ✓ | ✗ | ✓ | 64.0 | 76.6 | 90.0 |
| ✓ | ✓ | ✓ | ✓ | 64.0 | **76.8** | **90.2** |

Table 5: **Models used as teachers.** We use RealTracker online as a student model and ablate different combinations of teacher models. The first row corresponds to the model trained only on synthetic data. The second row corresponds to self-training. Generally, the more diverse teachers we have, the better is the tracking accuracy ($\delta_{\text{avg}}$).

## 4.2 SCALING EXPERIMENTS

In Figure 1, we show how RealTracker, LocoTrack, and CoTracker (Karaev et al., 2024b) improve with our pseudo-labeling pipeline as the training set size increases. Starting with models pre-trained on a synthetic dataset (Greff et al., 2022) (0 at x-axis), we train them on progressively larger real data sets: 0.1k, 1k, 5k, 10k, 30k, and 100k videos. Models are trained to convergence on their respective subsets. All models improve with just 0.1k real-world videos and continue improving with more. Improvements for RealTracker online, offline, and LocoTrack tend to plateau after 30k videos, likely because the student surpasses the teachers. This may also explain why CoTracker, initially much weaker than two of its teachers (RealTracker online and offline), keeps improving up to and possibly beyond 100k videos, which is the maximum we can afford to explore. Our training strategy is effective for all these models. We analyse the effect of using a scaled RealTracker as a new teacher in the supplement. For comparison, BootsTAPIR (Doersch et al., 2024) uses 15 million real videos and a complex protocol involving augmentations, loss masks, and more.

Interestingly, we found that training RealTracker with its own predictions as annotations without other teachers (i.e., self-training) further improves the results on all the TAP-Vid benchmarks by +1.2 points on average (see Table 4). Presumably, fine-tuning on real data, even with its own annotations, helps the model reduce the domain gap between real and synthetic data.

## 4.3 ABLATIONS

**Cross-track attention.** Table 3 shows that cross-track attention improves results, particularly for occluded points (+5.1 occluded vs. +1.6 visible on Dynamic Replica). This is because by using cross-track attention, the model can guess the positions of the occluded points based on the positions of the visible ones. This cannot be done if the points are tracked independently.

**Teacher models.** We assess the impact of using multiple teachers for generating pseudo-labels in Table 5. We start by removing weaker models and always keep the student model itself as a teacher. We demonstrate that removing a teacher always leads to worse results compared to the last row, where we train with all four teacher models. This shows that every teacher is important and that the student model can always extract complementary knowledge, even from weaker teachers.

**Point sampling.** In Table 6 we have explored alternative point sampling methods, including Light-Glue (Lindenberger et al., 2023), SuperPoint (DeTone et al., 2018), and DISK (Tyszkiewicz et al.,

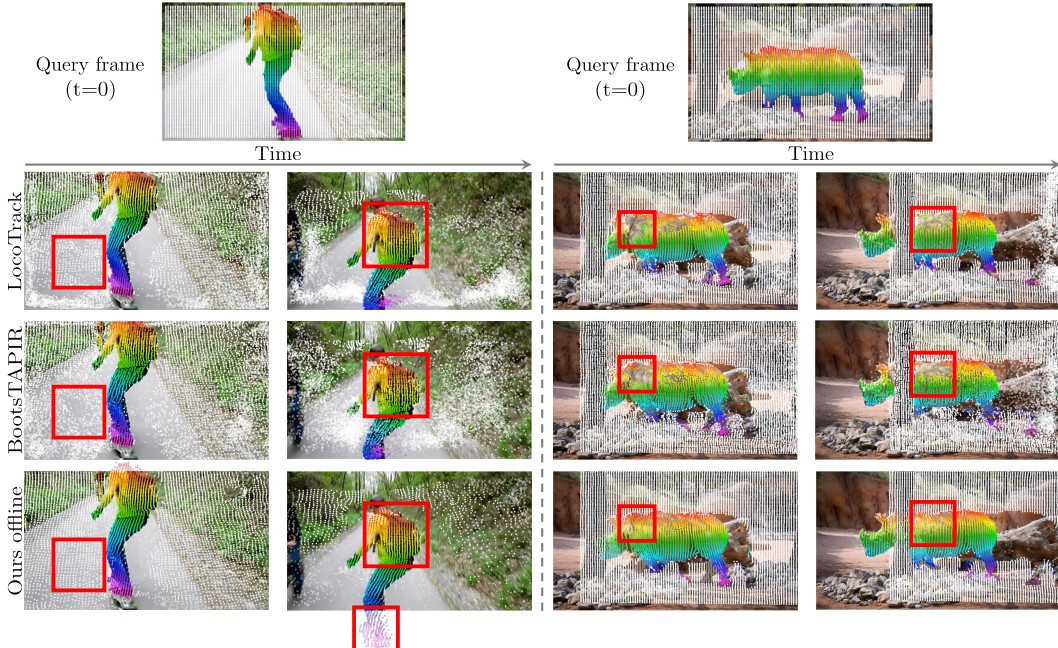

Figure 4: **Qualitative comparison.** Tracking a grid of $100 \times 100$ points from the first frame should maintain grid patterns in future frames when the motion is simple. LocoTrack and RealTracker are more consistent than BootsTAPIR, but neither LocoTrack nor BootsTAPIR can track through occlusions and also lose more background (1st column) and object points (3rd and 4th columns).

| Sampling | Kinetics | DAVIS | RoboTAP | RGB-S |
|---|---|---|---|---|
| Uniform | 67.9 | 76.9 | 78.4 | **84.0** |
| SuperPoint | 68.1 | 76.7 | **78.9** | 81.9 |
| DISK | 68.0 | 76.7 | 78.6 | 82.7 |
| SIFT | **68.2** | **77.0** | 78.8 | 83.3 |

| Frozen head | Average on TAP-Vid | | |
|---|---|---|---|
| | AJ ↑ | $\delta_{\text{avg}}$ ↑ | OA ↑ |
| ✗ | 63.2 | 76.6 | 86.3 |
| ✓ | **64.0** | **76.8** | **90.2** |

Table 6: **Point sampling strategies** on $\delta_{\text{avg}}$ on TAP-Vid. SIFT is overall best, but the method is robust w.r.t. this choice.

Table 7: Average AJ, $\delta_{\text{avg}}$ and OA on TAP-Vid, where **freezing** the confidence and visibility heads improves performance, **avoiding forgetting**.

2020). The choice of the sampling method does not significantly affect the performance. However, SIFT sampling results are consistently high across all the TAP-Vid datasets.

**Freezing the confidence and visibility head.** In Table 7, we show that splitting the transformer head into a separate head for tracks and a head for confidence and visibility helps to avoid forgetting when supervising only tracks while training on real data. We freeze the head for confidence and visibility at this stage. This improves AJ by $+0.8$ and OA by $+3.9$ on TAP-Vid on average.

## 5 CONCLUSION

We introduced RealTracker, a new point tracker that outperforms the state-of-the-art on TAP-Vid and other benchmarks. RealTracker's architecture combines several good ideas from recent trackers but eliminates unnecessary components and significantly simplifies others. RealTracker also shows the power of a simple pseudo-labelling training protocol, where real videos are annotated utilizing several off-the-shelf trackers and then used to fine-tune a model that outperforms all teachers. With this protocol, RealTracker can surpass trackers trained on $\times 1,000$ more videos. By tracking points jointly, RealTracker handles occlusions better than any other model, particularly when operated in offline mode. Our model can be used as a building block for tasks requiring motion estimation, such as 3D tracking, controlled video generation, or dynamic 3D reconstruction.

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

APPENDIX

Please refer to the **index.html** webpage in the supplement for a visual and animated comparison between methods.

## A    IMPLEMENTATION DETAILS

We pre-train both online and offline model versions on synthetic **TAP-Vid-Kubric** (Doersch et al., 2022; Greff et al., 2022) for 50,000 iterations on 32 NVIDIA A100 80GB GPUs with a batch size of 1 video. We train RealTracker online on videos of length $T = 64$ with a window size of 16, and sample 384 query points per video with a bias towards objects. Since the online version tracks only forward in time, we sample points primarily at the beginning of the video. We train the offline version on videos of length $T \in \{30, 31, \ldots, 60\}$ with time embeddings of size 60. We interpolate time embeddings to the current sequence length both at training and evaluation. We sample 512 query points per video uniformly in time. Both models are trained in bfloat16 with gradient norm clipping using PyTorch Lightning (Falcon & The PyTorch Lightning team, 2019) with PyTorch distributed data parallel (Li et al., 2020). The optimizer is AdamW (Loshchilov & Hutter, 2017) with $\beta_1 = 0.9$, $\beta_2 = 0.999$, learning rate $5 \cdot 10^{-4}$, and weight decay $1 \cdot 10^{-5}$. The optimizer adopts a linear warm-up for 1000 steps followed by a cosine learning rate scheduler.

We scale RealTracker on a dataset of Internet-like videos primarily featuring humans and animals. We visualize the scaling pipeline in Figure 5. To ensure the quality and relevance of our training data, we use caption-based filtering with specific keywords to select videos containing real-world content while excluding those with computer-generated imagery, animation, or natural phenomena that are challenging to track, such as fire, lights, and water.

When training on real data, we use a similar setup while reducing the learning rate to $5e - 5$ with the same cosine scheduler without warm-up. We train both online and offline versions for 10,000 iterations with 384 tracks per video sampled with SIFT on eight randomly selected frames with frame sampling biased towards the beginning of the video.

Following (Karaev et al., 2024b), when evaluating RealTracker online on TAP-Vid, we add 5×5 points sampled on a regular grid and 8×8 points sampled on a local grid around the query point to provide context to the tracker. We do the same for the scaled offline version during inference. The Kubric-trained offline version, however, relies on uniform point sampling during training. For this model, during evaluation on TAP-Vid, we instead sample 1000 additional support points uniformly over time.

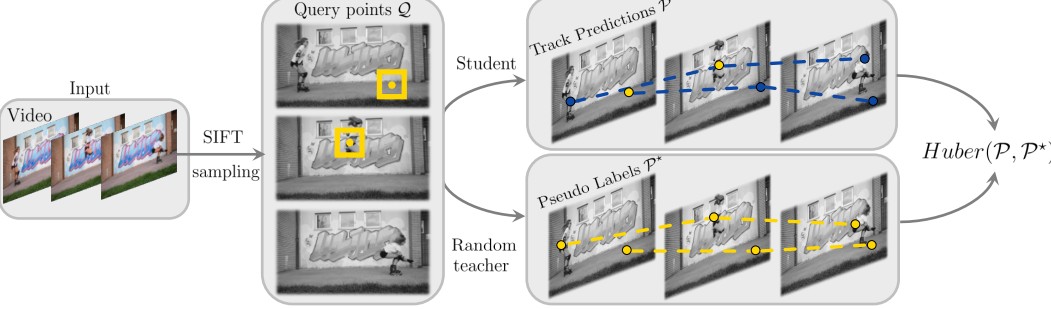

Figure 5: **Scaling pipeline**. Given a video, we randomly choose 8 frames and sample 384 query points across these frames using SIFT Lowe (1999). Then, we predict tracks for these query points with the student and randomly selected teacher models. Finally, we compute the difference between the predicted tracks and update the student model.

## B    PERFORMANCE

In Figure 6, we compare the speed of RealTracker with other point trackers. We measure the average time it takes for the method to process one frame, with the number of tracked points varying between

| Teacher selection strategy | Kinetics | DAVIS | RoboTAP | RGB-S |
|---|---|---|---|---|
| **Random** | **68.2** | **77.0** | **78.8** | **83.3** |
| Averaging | 67.4 | 76.5 | 77.9 | 82.4 |
| Median | 67.3 | 76.3 | 77.3 | 81.1 |

Table 8: **Supervision.** Random sampling of teachers consistently leads to better $\delta_{\text{avg}}$ on TAP-Vid compared to supervision with either the mean or the median of all teachers' predictions.

1 and 10,000. We average this across 20 videos of varying lengths from DAVIS. Even though Co-Tracker and RealTracker apply group attention between tracked points, the time complexity remains linear thanks to the proxy tokens introduced by (Karaev et al., 2024b). While all the trackers exhibit linear time complexity depending on the number of tracks, RealTracker is approximately 30% faster than LocoTrack (Cho et al., 2024), the fastest point tracker to date.

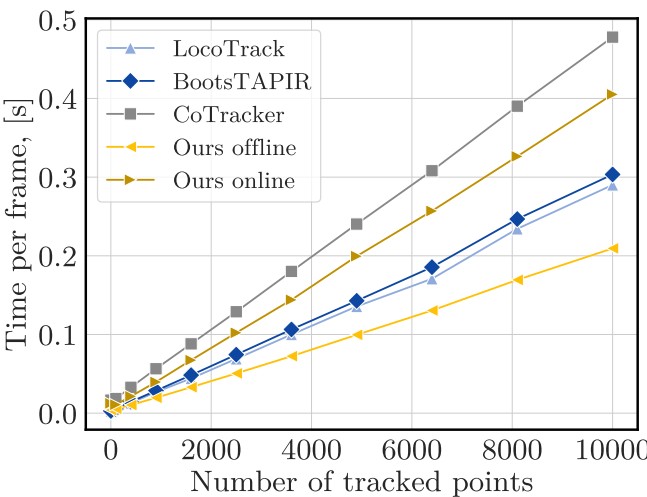

Figure 6: **Efficiency.** We evaluate the speed of different trackers on DAVIS depending on the number of tracks and report the average time each tracker takes to process a frame. Our offline architecture is the fastest among all these models, with LocoTrack being the fastest tracker to date.

## C  ADDITIONAL EXPERIMENTS

**Training with the average of teachers' predictions.**   Interestingly, we found that aggregating the predictions of multiple teachers instead of using a random teacher does not improve performance, as shown in Table 8, whereas incorporating additional teachers into training consistently enhances the quality of our student model, demonstrated in Table 5.

**Repeated scaling.**   We study the effect of iterative scaling to investigate the limits of our multi-teacher scaling pipeline. Specifically, we scale RealTracker offline using our pipeline, where one of the teachers is the model itself. We then take this trained student model and attempt to improve it further by re-applying the same scaling pipeline but with the original student model replaced by the newly trained student model as one of the teachers.

We find that this second round of scaling leads to slight improvements in performance metrics. This suggests that the student model has already distilled most of the knowledge from the other teachers during the initial training phase. We report the results in Table 9.

**Convergence behavior during scaling.**   We examine the convergence behavior of our scaling pipeline by fixing the dataset and all the hyper-parameters, varying only the number of iterations over the dataset. We show in Table 10 that increasing the number of iterations leads to improved performance on TAP-Vid, but with diminishing returns. Specifically, we observe a saturation point

| Model | Average on TAP-Vid | | |
|-------|------|---------------|------|
| | AJ ↑ | $\delta_{avg}$ ↑ | OA ↑ |
| Kub+15k | 64.0 | 76.8 | **90.2** |
| Kub+15k+15k | **64.2** | **76.9** | 89.7 |

Table 9: **Repeated scaling.** We scale Real-Tracker offline, then start from a scaled model, and scale it again with the scaled model as one of the teachers. Repeated scaling slightly improves tracking accuracy.

| Num. of iterations | Average on TAP-Vid | | |
|--------------------|------|---------------|------|
| | AJ ↑ | $\delta_{avg}$ ↑ | OA ↑ |
| 1k | 63.3 | 75.6 | 87.5 |
| **10k** | 64.0 | 76.8 | **90.2** |
| 30k | 64.4 | 76.8 | 89.7 |
| 60k | **64.4** | **77.0** | 89.5 |

Table 10: **Longer training on 10k videos.** We train RealTracker offline for longer to determine the optimal number of iterations for a given number of videos. As a trade-off between training costs and the results obtained, we use the same number of iterations as the number of videos.

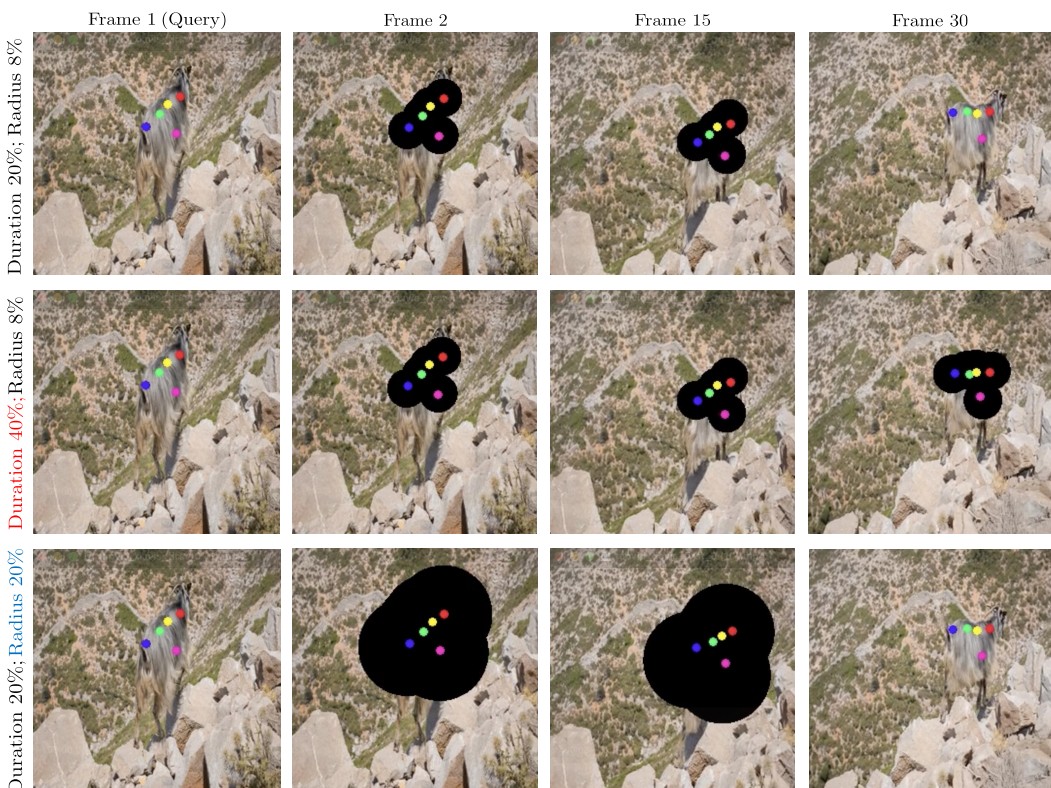

Figure 7: **Occlusions.** We occlude all tracked points with black circles of different sizes for several consecutive frames. We experiment with different scenarios, discussed in the text. For each, we also visualize the predicted positions of the tracked points.

beyond which further increases in the number of training iterations do not yield significant improvements in model quality. We thus use the same number of iterations as the number of training videos with a batch size of 32, iterating over each video 32 times.

**Occlusions.** We investigate the effect of occlusions of different sizes and lengths on the tracking accuracy on TAP-Vid DAVIS. Specifically, we occlude all the tracked points with black circles of different sizes for several consecutive frames (see Figure 7). We then measure how this affects the tracking accuracy using offline tracking in various scenarios, discussed next.

| Radius (% of img. width) $\delta_{avg}$ ↑ | |
|---|---|
| 0 | 76.9 |
| 4 | 48.8 |
| 8 | 42.2 |
| 12 | 39.8 |
| 20 | 36.4 |
| 40 | 30.2 |
| 80 | 23.3 |
| 100 | 19.9 |

| Duration (% of vid. len.) $\delta_{avg}$ ↑ | |
|---|---|
| 0 | 76.9 |
| 20 | 61.3 |
| 40 | 48.2 |
| 60 | 37.5 |
| 80 | 29.9 |
| 100 | 21.1 |

Table 11: **Varying size of occlusions.** We report tracking accuracy on DAVIS depending on the radius of artificially added occluding circles, which cover all tracked points for half of the video (30 frames on average, see Figure 7).

Table 12: **Varying duration of occlusions.** We report tracking accuracy on DAVIS depending on the duration of artificially added occluding circles with radius of 8% of the image width. Occluders cover all the tracked points, see Figure 7.

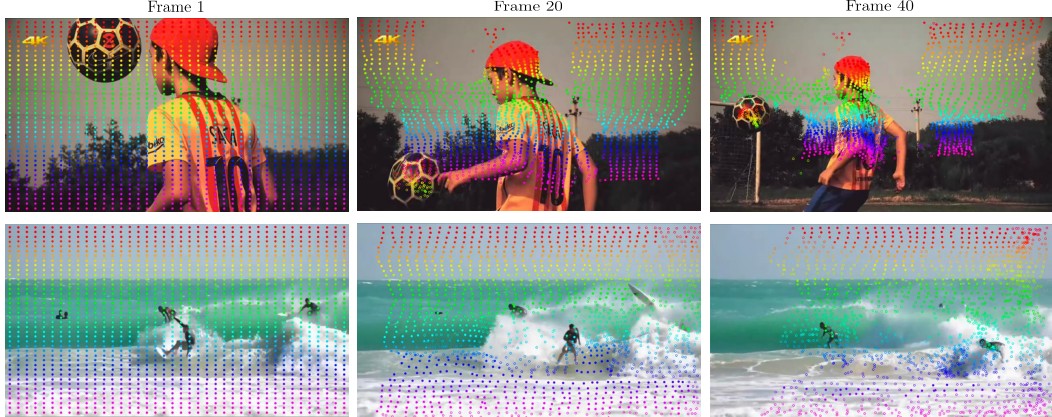

Figure 8: **Failure cases.** Featureless surfaces is a common mode of failure: the model cannot track points sampled in the sky or on the surface of water.

First, we show that RealTracker can successfully utilize the temporal context to track points through occlusions. In order to do so, we occlude all tracked points in each video for half of the video length, starting right after the query frame, but let the points be visible in the second half. The occluding circle is centered in the ground truth track. We vary the radius of the occluding circle and increase it from 0% to 100% of the video width. As Table 11 shows, the tracking accuracy is still 19.9% even with a radius of 100%; this is because the model sees the second half of the video and can track points there. If we occlude all rather than half the frames with a radius of 100%, the accuracy drops to 2%.

Second, we show that RealTracker can also successfully utilize the spatial context given by other tracked points. In order to do so, we fix the radius of the occlusion to 8% of the image width and vary the duration of the occlusion from 0 to 100% of the video length, with average video length being 60 frames. In Table 12, short occlusions of 20% of video length (12 frames on average) affect performance, but not significantly: accuracy drops from 76.9% to 61.3%. When occluding points for the whole duration of the video (100%), the model can still approximate the location of these points due to the presence of unoccluded support points (21.1% accuracy vs 2% when all points are occluded).

## D FAILURE CASES

In Figure 8, we show examples of failure cases. We track a grid of 40*40 points from the first frame and demonstrate that the model can not reliably track points sampled in the sky or on the surface of water, partly because the task is ambiguous in these cases: it is unclear whether the tracked point in

the sky should remain static or move with the camera. Other common sources of failure are tracking shadows of objects and tracking through long occlusions.

## E    LIMITATIONS

A key limitation of our pseudo-labeling pipeline is its reliance on the quality and diversity of teacher models. The observed saturation in performance on TAP-Vid during scaling suggests that the student model absorbs knowledge from all the teachers and, after a certain point, struggles to improve further. Thus, we need stronger or more diverse teacher models to achieve additional gains for the student model.

