# OpenReview forum: "RealTracker: Simpler and Better Point Tracking by Pseudo-Labelling Real Videos"
_ICLR.cc/2025/Conference — Submitted to ICLR 2025_

### Official Review · Reviewer_4EZJ · 2024-11-02

**Soundness:** 3
**Presentation:** 3
**Contribution:** 3
**Rating:** 5
**Confidence:** 4

**Summary:**

This paper introduces the RealTracker, a point tracker that combines several ideas from other related trackers but eliminates some components and simplifies others. RealTracker also designes a semi-supervised training protocol, where real videos are annotated utilizing several off-the-shelf trackers. With this protocol, RealTracker can achieve encouraging results on the Kinetics, RGB-S, and DAVIS datasets.

**Strengths:**

1. RealTracker combines valuable ideas from the recent state-of-the-art point trackers and eliminates some unimportant modules.
2. RealTracker proposes a simple semi-supervised training protocol and achieves better results on several public datasets compared to state-of-the-art trackers.
3. RealTracker explores the training scaling low via its proposed training protocol.

**Weaknesses:**

1. The idea of using trackers to annotate unlabeled datasets, such as [1], is not new.
2. The authors should use the Kub+15M data to train the CoTracker and TAPTR and verify the proposed method's effectiveness.
3. To prove the effectiveness of the RealTracker, it is suggested that confidence and visibility be visualized.
4. More ablation studies are suggested to verify that eliminating some modules in the listed trackers and simplifying some modules is useful, including the computation cost and tracking performance.

[1] Muller M, Bibi A, Giancola S, et al. Trackingnet: A large-scale dataset and benchmark for object tracking in the wild[C]//Proceedings of the European conference on computer vision (ECCV). 2018: 300-317.

**Questions:**

Please follow the weakness. If the issues are addressed,  I will improve the rating.

---

> ### Author Response · Authors · 2024-11-25
>
> > The idea of using trackers to annotate unlabeled datasets, such as [1], is not new.
>
> Thank you, we have cited [1] TrackingNet by Muller et al. in the revised version. Point tracking however is different from object tracking in an important way: it requires not only localizing the same object in another frame, but also **a specific point on the same object**, which makes the task more difficult.
> Here we work with point tracking and show that synthetic-trained teacher trackers **don't have to be better** than the student tracker to improve the student. For example, **synthetic-trained RealTracker is better than all its synthetic-trained teachers** and its performance still improves significantly after training with our pipeline.
>
>
> > The authors should use the Kub+15M data to train the CoTracker and TAPTR and verify the proposed method's effectiveness.
>
> The 15M dataset (15M from Kub+15M) introduced in BootsTAPIR was not released and nobody has access to this data. However, with our method, we improve over BootsTAPIR with much less data: RealTracker is better after training on just 15 **thousand** real videos compared to 15 **million** in BootsTAPIR.
> In Fig. 1 we show how CoTracker and LocoTrack improve after training with our pipeline on up to 100 thousand real videos. We don't have enough data and resources to scale training any further (to the proposed 15 million videos as was done in BootsTAPIR).
>
> > To prove the effectiveness of the RealTracker, it is suggested that confidence and visibility be visualized.
>
> We visualize visibility in Fig. 4 and in the supplementary videos (`index.html`, 5 videos in the header and videos from "Object-centric tracking on a regular grid"), where visible points are filled with color, and invisible points are left empty.
> Confidence simply estimates whether the point is within 12 pixels from the ground truth and allows to improve visibility predictions by discarding points with low confidence. So, confidence is already a part of the visualized visibility: `final_visibility = (visibility * confidence) > 0.6`
>
> > More ablation studies are suggested to verify that eliminating some modules in the listed trackers and simplifying some modules is useful, including the computation cost and tracking performance.
>
> Thank you for pointing this out. Below, we show ablations for re-adding the removed 4D correlation encoder or global matching. We ablated the online model, showing the average performance on TAP-Vid. Speed is measured in frames per second for the simultaneous tracking of 100 points:
>
> |	| AJ↑	| $\delta_\mathrm{avg}$↑| 	OA↑	| Speed↑ | Number of parameters↓ |
> | :---: |:---: | :---: | :---: | :---: | :---: |
> | RealTracker	| 64.0	| 76.8	| **90.2**	| **90 fps** | **25M**  |
> | RealTracker + 4D corr encoder	| **64.3**	| 76.9 | 90.1 |	63 fps | 28M |
> | RealTracker + Global matching	| 63.6	| **77.0** | 89.9	| 82 fps | 26M |
>
> The impact of these modules on performance is small and inconsistent across metrics, while the affect the speed of the method. We thus find that they are not necessary to be included in RealTracker.
>
> We are open to other suggestions for interesting experiment from the reviewer.

---

### Official Review · Reviewer_MNzP · 2024-11-03

**Soundness:** 3
**Presentation:** 3
**Contribution:** 2
**Rating:** 6
**Confidence:** 5

**Summary:**

the approach leverages other point trackers to produce training data  for their point tracker.  Supposedly less additional training data is required compared to other point trackers.   The biggest contribution is that the other trackers use real data and not synthetic data for training.   Other approaches in the past have typically used point data for tracking.

**Strengths:**

The paper incrementally builds upon point trackers by producing a better approach that leverages other point trackers to produce supervised training data.  In the past other trackers have used synthesized data however this is all based on real data.  The results seem to better than other point trackers.

**Weaknesses:**

It is not clear on what type of motions were tested, if parallax for motion is required, what about zooming like motions with no parallax, does the method work.
What % of occlusion in terms of coverage of the object and in terms of time occluded were not clearly tested.
The limitations and failure cases of the algorithm were not explored.

**Questions:**

From Table 2, it appears that the training set does matter in the results, The methods training with Kub+15M performed on average better than the methods trained with Kub, please explain and elaborate.  What is the difference?
Why does the offline method perform better than the online method, Intuitively I would assume the opposite?
What are the limitations and failure cases?
Table 6, why does SIFT turn on the best results?

---

> ### Author Response · Authors · 2024-11-25
>
> > It is not clear on what type of motions were tested, if parallax for motion is required, what about zooming like motions with no parallax, does the method work. What % of occlusion in terms of coverage of the object and in terms of time occluded were not clearly tested.
>
> The method works for any types of motions, including zooming motions.
> Thank you, we've conducted two additional experiments with occlusions on TAP-Vid DAVIS for the rebuttal. In both of these experiments, we investigate the effect of occlusions of different sizes and lengths on the tracking accuracy on TAP-Vid DAVIS. Specifically, we occlude all tracked points with black circles of different sizes for several consecutive frames and measure how this affects tracking accuracy of RealTracker offline while tracking query points (See appendix of the revised version of the paper for a visual explanation.)
>
> In the first experiment, we occlude all tracked points in each video for half of the video length, starting right after the query frame. The occluding circle is centered in the ground truth track. We vary the radius of the occluding circle and increase it from 0% to 100% of the video width. The reason why tracking accuracy is still 19.9% with a radius of 100\% is that the model sees the second half of the video and can track points there. If we occlude all the frames with a radius of 100%, the accuracy drops to 2%.
>
> | occlusion radius in % of image width | $\delta_\mathrm{avg}$↑ |
> | :---: |:---: |
> | 0 | 76.8 |
> | 4| 48.8 |
> | 8| 42.2 |
> | 12| 39.8 |
> | 20| 36.4 |
> | 40| 30.2 |
> | 80| 23.3 |
> | 100| 19.9 |
>
> In the second experiment, we fix the radius to 8\% of the image width and vary the duration of the occlusion from 0 to 100\% of the video length, with average video length being 60 frames. Short occlusions of 20\% of video length (12 frames on average) affect performance, but not significantly: accuracy drops from 76.9\% to 61.3\%. When occluding points for the whole duration of the video (100\%), the model can still somewhat predict where these points are thanks to supporting grid points.
> | occlusion duration (frames) | $\delta_\mathrm{avg}$↑ |
> | :---: |:---: |
> | 0 | 76.8 |
> | 20| 61.3 |
> | 40| 48.2 |
> | 60| 37.5 |
> | 80| 29.9 |
> | 100| 21.1 |
>
>
>
> > From Table 2, it appears that the training set does matter in the results, The methods training with Kub+15M performed on average better than the methods trained with Kub, please explain and elaborate. What is the difference?
>
> Kubric (Kub) is a synthetic dataset. Kub + 15M means synthetic pre-training with fine-tuning on 15 **million** real videos. Kub + 15k is synthetic pre-training and finetuning on 15 **thousand** real videos. In this table we show that fine-tuning improves the results over just synthetic pretraining, and that RealTracker is better after training on 15 **thousand** videos compared to BootsTAPIR trained on 15 **million** videos. BootsTAPIR did not release their 15M training video dataset so it is not reproducible.
>
> > Why does the offline method perform better than the online method, Intuitively I would assume the opposite?
>
> The offline method has access to all the video frames at once (more context than the online method that does sequential processing), so the offline method can deal better with occlusions. The offline method is able to reason across the whole video and thus can track points forward and backward in time, while the online version operates in a sliding-window manner and can track only forwards.
>
> > What are the limitations and failure cases?
>
> We discuss limitations of the proposed method in the appendix of the paper:
> A key limitation of our pseudo-labeling pipeline is its reliance on the quality and diversity of teacher
> models. The observed saturation in performance on TAP-Vid during scaling suggests that the student model absorbs knowledge from all the teachers and, after a certain point, struggles to improve further. Thus, we need stronger or more diverse teacher models to achieve additional gains for the student model.
>
> Thank you, we've now included failure cases in the revised version of the paper. Please see the revised version for visuals.
> Featureless surfaces is a common mode of failure: the model cannot track points sampled in the sky or on the surface of water. Other common sources of failure are tracking shadows of objects and tracking through long occlusions.
>
> > Table 6, why does SIFT turn on the best results?
>
> Sift is just slightly better. The model is more or less indifferent to the choice of point sampling.

---

### Official Review · Reviewer_PQwv · 2024-11-06

**Soundness:** 3
**Presentation:** 3
**Contribution:** 2
**Rating:** 5
**Confidence:** 3

**Summary:**

1. The authors address the redundancy in modules of various existing point tracking models and propose RealTracker, a network with simplified architecture that achieves better performance and faster processing speed.

2. The authors leverage existing models to generate pseudo-labels for real video data, enabling effective utilization of unlabeled videos for network fine-tuning, which further enhances performance.

3. The authors analyze the impact of real data scale on the network model's performance, providing insights into the relationship between dataset size and tracking effectiveness.

**Strengths:**

1. The paper's motivation is well-justified, particularly in its approach to eliminate model redundancies, resulting in a more lightweight yet powerful architecture.
2. The paper demonstrates effective utilization of unlabeled real-world datasets for training, achieving significant performance improvements through this approach.
3. The experimental analysis is comprehensive, and the visualization results are particularly impressive in demonstrating the model's capabilities.

**Weaknesses:**

1. The methodology appears to be more engineering-oriented rather than theoretically innovative, primarily consisting of combinations and modifications of existing methods. The pseudo-label fine-tuning approach is relatively common. Given this is a **deep learning conference**, the technical contributions seem somewhat limited.
2. As acknowledged in the limitations section, the model's improvement of performance is heavily dependent on the teacher model's capabilities. This strong reliance on existing methods' performance creates a ceiling effect where the training results are constrained by the teacher model's performance limits, potentially reducing the method's generalizability.
3. The authors aim to bridge the domain gap using real-world dataset training. However, the paper lacks substantial technical innovation in terms of cross-domain adaptation techniques. The approach merely relies on real-data fine-tuning and teacher model voting effects for enhanced robustness, neither of which represents a significant contribution to the field of domain adaptation. More sophisticated cross-domain strategies or novel technical approaches would have strengthened the paper's contribution in addressing the domain gap problem.

**Questions:**

1. The terminology "self-supervised fine-tuning" is indeed questionable in this context. Using state-of-the-art models from the same domain to generate pseudo-labels for supervision is more aligned with teacher-student learning or pseudo-labeling approaches rather than traditional self-supervised learning, where the supervision signals are typically derived from the data itself without external models.

2. The incorporation of domain adaptation strategies during the fine-tuning process would have significantly enhanced the paper's contribution. This could have included techniques specifically designed to address domain shift and better align feature distributions between source and target domains.

---

> ### Author Response · Authors · 2024-11-25
>
> > The methodology appears to be more engineering-oriented rather than theoretically innovative. The pseudo-label fine-tuning approach is relatively common. The technical contributions seem somewhat limited.
>
> RealTracker is the first paper to analyse scaling effects of training point trackers on pseudo labels (see Fig. 1). In this paper, we show that it is possible to improve the performance of any synthetic-trained point tracker on real data using only other **synthetic-trained** teachers. Moreover, these teacher trackers **don't have to be better** than the student tracker. For example, **synthetic-trained RealTracker is better than all its synthetic-trained teachers** and its performance still improves significantly after training with our pipeline.
>
> > The model's improvement of performance is heavily dependent on the teacher model's capabilities. This strong reliance on existing methods' performance creates a ceiling effect where the training results are constrained by the teacher model's performance limits, potentially reducing the method's generalizability.
>
> The model improves with more teachers even if they are all trained on the same dataset (Kubric) and even if all of them are worse than the student model (Tab. 5). So, the student model is not bounded by their performance, it just saturates after absorbing knowledge from training with different teachers for a while. This allows to improve any existing point tracker, even if it is already better than all the teacher models (for example, RealTracker, Tab. 5). In Fig. 1, we show the universality of the proposed pipeline: it improves other point trackers, such as LocoTrack and CoTracker. Interestingly, using only the model itself as a teacher also improves the results (Tab. 4).
>
> > The paper lacks substantial technical innovation in terms of cross-domain adaptation techniques. The approach merely relies on real-data fine-tuning and teacher model voting effects for enhanced robustness, neither of which represents a significant contribution to the field of domain adaptation.
>
> This is the first paper to systematically explore self-training on real data for point tracking. The state-of-the-art point tracker BootsTAPIR is trained on **15 million** real videos. Here we show that it is possible to outperform it by training only on **15 thousand** real videos with a simpler training protocol.
> The fact that we propose a simple method to do so is, in our view, a feature: we set a strong baseline for others to build on. In addition, we show for the first time how these trackers scale with the amount of training data, which is in its own right an important empirical analysis that will guide future research. We also obtain a result which is practically useful and important. Just like many are building on trackers like CoTracker2, we expect that many will take advantage of our new tracker. Besides, we also provide a substantially better tracker architecture, which is not only state-of-the-art, but faster and simpler that those it superseeds. We believe that this will also be of great interest to the community.
>
> > The terminology "self-supervised fine-tuning" is indeed questionable in this context.
>
> We agree that this method is more aligned with pseudo-labelling approaches, we have replaced all mentions of "self-supervised training" with "pseudo-labelling" in the revised version.

---

### Official Review · Reviewer_Gpaf · 2024-11-09

**Soundness:** 3
**Presentation:** 3
**Contribution:** 3
**Rating:** 6
**Confidence:** 4

**Summary:**

This paper proposes a simpler and better point tracking approach by pseudo-learning real videos. Specifically, the proposed approach allows real videos without annotations to be used during training by generating pseudo-labels using off-the-shelf teachers. The proposed approach explores to use real video for training point tracking models w/o annotations. Moreover, the authors also study the scaling law to understand the impact of using more real training videos.

**Strengths:**

- The paper focuses on an interesting problem in the community, i.e., aiming to explore to train TAP models w/ real videos w/o annotations, since the previous approaches mainly focus on learning w/ synthetic datasets;
- The proposed RealTracker shows that a simpler architecture and training protocols can outperform SOTA trackers like BootsTAPIR and LocoTrack;
- The paper is well written and organized;

**Weaknesses:**

- Using pseudo-labels for training trackers is well explored, e.g., for some online learning-based trackers like Dino-Tracker, it uses pre-computed optical flow which provides the pseudo ground truth pixel-level correspondences for online training the tracker. For DinoTracker3, pseudo-labelling is explored. Please illustrate more differences with these trackers for better highlighting the contributions;
- Are there any specific concerns for choosing a teacher model for pseudo label generation? Does the better teacher model with higher tracking performance commonly lead to better tracking performance? Can a single teacher model well support the tracker learning?
-  In Table 2, the time of the per frame and per tracked point is shown. For the online variant, what’s the overall tracking speed (i.e., fps) given an online testing video?
- Missing Refs for discussion. For completeness, please include more pseudo-label based tracker training approaches [1,2,3,4] for discussion in the related work.

[1] Progressive Unsupervised Learning for Visual Object Tracking;

[2] Unsupervised Learning of Accurate Siamese Tracking;

[2] DINO-Tracker: Taming DINO for Self-Supervised Point Tracking in a Single Video;

[3] CoTracker3: Simpler and Better Point Tracking by Pseudo-Labelling Real Videos;

**Questions:**

Overall, I think this is an interesting paper that focuses on an essential problem in the community, i.g., enabling existing TAP trackers to leverage real videos w/o annotations for training. The idea is somewhat incremental but effectively addresses an essential problem in a simple yet effective way. Thus my current rating is ``accept''. I would like to see more author rebuttal in terms of differences w/ existing pseudo label based approaches as mentioned above.

---

> ### Author Response · Authors · 2024-11-25
>
> > Using pseudo-labels for training trackers is well explored, e.g., Dino-Tracker with precomputed flow, CoTracker3 with pseudo-labelling. Please illustrate more differences with these trackers for better highlighting the contributions.
>
> DinoTracker is a test-time optimization method, which means it needs to be optimized for each video during inference. Fitting DinoTracker to a single video with 100 frames takes about 1.6 hours. In contrast, we pretrain a network on a real dataset using pseudo labels and then evaluate it on five different benchmarks without additional test-time optimization. RealTracker operates in real time (90 frames per second for 100 points). Also, optical flow is one out of five losses used for test-time optimization in DinoTracker. Here we supervise with a single loss for pseudo labelled tracks. CoTracker3 was released after the ICLR deadline.
>
>
> > Are there any specific concerns for choosing a teacher model for pseudo label generation? Does the better teacher model commonly lead to better tracking performance? Can a single teacher model well support the tracker learning?
>
> In Tab. 5 we ablate different teacher setups and show that adding any teacher improves performance, even if the teacher itself is worse than the student model. The more diverse the teachers, the better the result. It appears that the student model absorbs complementary knowledge from different teachers even if they're worse than the student model. A better teacher model indeed leads to better tracking performance (adding CoTracker or another RealTracker as a teacher is better than adding TAPIR). Interestingly, a single teacher (the model itself) also improves the tracking results (see Tab. 4)!
>
> > In Table 2, the time of the per frame and per tracked point is shown. For the online variant, what’s the overall tracking speed (i.e., fps) given an online testing video?
>
> The speed of the online model is 25 frames per second for 1000 simultaneously tracked points, 90 frames per second for 100 points.
>
>
> > Missing Refs for discussion. For completeness, please include more pseudo-label based tracker training approaches [1,2,3,4] for discussion in the related work.
>
> Thank you for pointing this out, we have cited these works in the revised version of the paper:
>
>
> Progressive Unsupervised Learning for Visual Object Tracking[1] introduced an unsupervised learning framework that entirely removed the need for annotated videos in visual tracking.
>
>
> Unsupervised Learning of Accurate Siamese Tracking[2] proposed an unsupervised learning framework based on siamese networks for training trackers with cycle consistency.
>
> DINO-Tracker[3] combines test-time per-video optimization with DINO features to improve point tracking.
>
> CoTracker3[4] introduces a more efficient architecture and a simple pseudo-labelling pipeline to further improve its performance by training on real data.

---

### Author Response · Authors · 2024-11-25

We thank all reviewers for their thoughtful feedback. The reviewers find that RealTracker [R1]`is an interesting paper that focuses on an essential problem in the community` with a [R2]`well-justified motivation`. RealTracker's  [R2] `visualization results are particularly impressive in demonstrating the model's capabilities`, it [R3,R4] `achieves better results on several public datasets compared to state-of-the-art trackers`. We address reviewers' comments below and have already incorporated their feedback in the uploaded revised version.

---

### Meta-Review · Area_Chair_xtyL · 2024-12-21

**Metareview:**

The article has received feedback from four reviewers, to which the authors have provided partial responses. However, the lack of intense discussion among the reviewers suggests that the paper may have limited novelty and topicality. It is hoped that the authors will take the reviewers' comments into consideration for further refinement of the manuscript and wish them success in future submissions. The current version of the paper will be rejected.

**Additional Comments On Reviewer Discussion:**

1. The task setting of the article is not particularly novel; as the reviewers have pointed out, a similar design was employed by TrackingNet.
2. The experimental validation section remains insufficient and calls for further optimization.

---

### Decision · Program_Chairs · 2025-01-22

Reject